# Histological Evaluation of Resected Tissue as a Predictor of Survival in Horses with Strangulating Small Intestinal Disease

**DOI:** 10.3390/ani13172715

**Published:** 2023-08-26

**Authors:** David Bardell, Guido Rocchigiani, Lorenzo Ressel, Peter Milner

**Affiliations:** 1Department of Equine Clinical Sciences, Institute of Infection, Veterinary and Ecological Sciences, University of Liverpool, Leahurst Campus, Chester High Road, Neston CH64 7TE, UK; pmilner@liverpool.ac.uk; 2Department of Veterinary Anatomy Physiology and Pathology, Institute of Infection, Veterinary and Ecological Sciences, University of Liverpool, Leahurst Campus, Chester High Road, Neston CH64 7TE, UK; guido.rocchigiani@liverpool.ac.uk (G.R.); ressel@liverpool.ac.uk (L.R.)

**Keywords:** equine, colic, prognosis, ischaemic bowel disease, intestine, pathology

## Abstract

**Simple Summary:**

Equine colic is a serious and potentially life-threatening clinical problem. This is particularly the case when the condition is due to strangulation of the small intestine, which requires part of it to be surgically removed (resected). Failure to re-establish functional motility post-operatively is a common reason for post-operative euthanasia, even if surgery has been successfully accomplished. The margins for a resection are typically determined by gross appearance of the small intestine combined with appropriate preservation of vascular supply to the remaining tissue. We hypothesized that histological evaluation of resected intestine could highlight changes indicative of unhealthy gut tissue left in situ, which could predict the likelihood of continued intestinal dysfunction and therefore survival. We graded the histological appearance of grossly normal and abnormal tissue resected from horses with strangulating small intestinal lesions and control tissue taken from horses euthanised for reasons unrelated to intestinal disease. There was no difference between the histological scores of grossly looking normal regions of the resected tissue and control tissues, nor was this associated with survival post-surgery. Grossly abnormal resected tissue was significantly different to the control tissue, and those horses in which the histological appearance showed greatest tissue disruption proximally (orally) demonstrated longer survival times.

**Abstract:**

Strangulating small intestinal disease (SSID) in horses carries a poor prognosis for survival, especially following resection of ischaemic tissue. The margins of a resection are principally based on visual appraisal of the intestine during surgery. We hypothesized that histological evaluation of resected tissue may identify occult changes indicative of prognosis. Small intestinal samples from 18 horses undergoing resection for SSID and 9 horses euthanised for reasons unrelated to gastrointestinal pathology were utilised. Histological appearance was used to generate a ‘total damage score’ (TDS) for the control tissue, grossly normal tissue at oral and aboral extremities (sections OR1 and AB1) of the resected intestine, and oral and aboral extremities of visually abnormal tissue (sections OR2 and AB2) from SSID horses. The relationship between TDS and long-term post-operative survival was investigated. TDS was not different between control tissues and OR1 and AB1 sections. Five surgical cases were alive at follow-up, the longest follow-up time being 2561 days. Based on the median scores for SSID cases versus controls, cut-off values were generated to evaluate post-operative survival versus TDS. Only OR2 TDS was significantly associated with survival, with a higher (worse) score indicating longer survival. More severe tissue insult may expedite rapid progression to surgery, improving post-operative outcomes.

## 1. Introduction

Colic in horses is a serious and potentially life-threatening clinical challenge. Whilst both the aetiology and anatomical location of the pathological processes can be very diverse, colic due to small intestinal obstruction is associated with significantly poorer survival than that due to caecal or large intestinal obstruction [1,2]. The likelihood of survival is further decreased with strangulating rather than simple obstruction, and if resection following ischaemic insult is required [3,4]. If resection of the compromised small intestine is deemed necessary, the oral and aboral margins of tissue bracketing the devitalized intestine are determined largely by subjective evaluation of the gross appearance and surgeon experience, in conjunction with ensuring appropriate preservation of mesenteric vasculature to maintain effective perfusion of the intestine to be left in situ. Post-operatively, mortality is most likely to occur in the first 7–10 days, and is frequently consequent to recurrence of colic signs, post-operative ileus and cardiovascular instability, consistent with endotoxaemic shock [5,6,7]. Despite successful surgery, failure to re-establish normal propulsive motile function in the early post-operative period can result in the destruction of the horse. Histological changes associated with ischaemia reperfusion injury of the small intestine have been investigated experimentally in several species, including mice [8], rats [9], dogs [10], pigs [11], man [12] and horses [13,14,15,16,17]. In horses, histological changes in tissue resected from clinical cases of strangulating small intestinal disease (SSID) have been described [18,19,20], but information on correlation of these with outcomes is sparse. Meschter et al. (1986) included 30 horses with a range of primary lesions, 9 of which were classified as SSID. Of these, only one survived, but the duration of survival/non-survival was not stated [18]. Gerard et al. (1999) only evaluated the mucosal and serosal layers, and reported that six out of nine horses were alive two years later [19]. De Ceulaer et al. (2011) reported morphological changes in mucosal and muscularis layers in 18 horses with SSID, but provided no information on the outcomes [20]. We hypothesized that histological evaluation of the margins of resected small intestine sections may identify occult changes which could act as prognostic indicators for survival following surgery. This study aimed to investigate if the presence or severity of degenerative changes identified histologically in resected small intestinal tissue could provide a useful mechanism for predicting post-operative survival in horses suffering from SSID. The objective was to determine if the tissue architecture of full-thickness samples from oral and aboral margins and regions of transition from grossly normal to abnormal appearance of small intestine resected from horses with SSID differed from that of control tissue obtained from horses without gastrointestinal disease. Survival analysis was then performed to investigate the relationship between histological grading and post-operative survival.

## 2. Materials and Methods

This case control study was conducted over a two-year period, following institutional ethical approval (RETH000689; VREC219a). Horses admitted to the University of Liverpool Philip Leverhulme Equine Hospital for investigation of acute abdominal disease and taken to surgery with a presumptive diagnosis of small intestinal obstruction were considered eligible for inclusion. Clinical data collected at presentation included heart rate (bpm), total plasma protein (g/L), packed cell volume (PCV, %) and peripheral blood lactate (mmol/L). If small intestinal strangulation requiring resection was identified at surgery, these horses were included in the study and tissue samples collected. The contents of the affected intestinal segment were evacuated aborally, and bowel clamps were applied orally and aborally to delineate the section for removal. Mesenteric vessels were then ligated and transected prior to removal of the intestinal segment by sharp dissection. The resected segment was then measured (cm) and photographed, and five-millimetre-thick, full-circumference, cross-sectional slices taken from four locations:Approximately 1 cm from the oral and aboral extremities (OR1 and AB1, respectively), avoiding tissue which had been compressed from the application of intestinal clamps, to represent the condition of tissue left in situ as closely as possible.At the regions of most visually obvious change from normal to abnormal tissue (the transition zone), samples were taken of the tissue of the most abnormal appearance adjacent to the transition zones at both oral and aboral ends (OR2 and AB2, respectively) (Figure 1).

Ileal involvement (yes/no) and type of anastomosis performed (jejuno-jejunal, jejuno-ileal and jejuno-caecal) were also recorded.

Control tissue was obtained from horses euthanised for conditions unrelated to the gastrointestinal system. Euthanasia was performed via an injection of a combination of quinalbarbitone and cinchocaine (Somulose^®^, Dechra, UK), following placement of a jugular venous cannula specifically for this purpose. Once death had been confirmed, a ventral midline incision was made, a short length of the small intestine exteriorized and a section, approximately five centimetres long, removed, from which a five-millimetre-thick, full-circumference, cross-sectional slice was taken for histological analysis.

The tissue slices were immediately placed into 10% formalin solution for 24 h, before being transferred into tissue cassettes. Formalin fixation, dehydration and infiltration were then performed in a Tissue-Tek vacuum infiltration automatic tissue processor using the following protocol:10% formalin for 1 h 45 min;70% ethanol for 30 min;70% ethanol for 30 min;86% ethanol for 30 min;96% ethanol for 1 h 30 min;Absolute ethanol for 1 h 30 min;Absolute ethanol for 1 h 30 min;Xylene for 1 h;Xylene for 2 h;Wax 1 for 1 h;Wax 2 for 1 h;Wax 3 for 1 h;Wax 4 for 1 h.

Paraffin embedded samples were then cut into 4 µm thick sections using a Leica RM2125 RT microtome, placed on slides, dewaxed, rehydrated, stained using haematoxylin and eosin and mounted in DPX.

Sections were examined at 10× objective magnification by light microscopy, and histological appearance was graded according to the scheme given in Table 1. Examination and grading were performed blindly by a single operator (G.R.) who was unaware of the origin of specimens and clinical outcomes of the cases. Representative images are shown in Figure 2, Figure 3, Figure 4 and Figure 5. The grades for each mural layer were then summed to produce a total damage score for each intestinal section.

Follow-up data on clinical outcomes from surgical cases were obtained either from the hospital clinical management software programme (Tristan Veterinary Software (version 1.8.3.1110), Aberdeen, UK) or by telephone contact with the veterinary surgeon or practice responsible for the initial referral, to determine if the horse was still alive. If the horse had been euthanised following discharge from hospital, we ascertained whether euthanasia had been related to colic symptoms or an unrelated reason.

### Statistical Analysis

Data were analysed using SPSS Statistics for Windows, Version 27 (IBM Corp. Armonk, NY, USA). Normality of data was checked using the Shapiro–Wilk test. Non-normally distributed data (peripheral lactate) were log-transformed.

Chi-squared analysis was used to explore differences in histological total damage scores between small intestinal samples from the control horses and regions OR1, OR2, AB1 and AB2 of the resected tissue from horses undergoing surgery for correction of small intestinal strangulation. The Mann–Whitney U test was used to investigate differences in the total damage scores for regions OR1, OR2, AB1 and AB2 between horses which had died or were euthanised for colic-related reasons, and those still alive at follow-up. The two-sided Fisher’s Exact Test was used to investigate whether the presence or absence of thrombotic vessels or serosal fibrin in regions OR1, OR2, AB1 and AB2 was associated with post-operative survival.

Univariable data analyses (Chi-squared analysis for categorical data and Student’s *t*-test for continuous data) were conducted for each explanatory variable (sex; horse type (Cob versus non-Cob); heart rate (bpm); total plasma protein (g/L); packed cell volume (PCV, %) and log peripheral blood lactate (mmol/L) at the time of admission; histological score; serosal fibrin (yes/no) and thrombosis (yes/no) at sites OR1, OR2, AB1 and AB2; length resected (cm); ileal involvement (yes/no); and anastomosis type (jejuno-jejunal, jejuno-ileal and jejuno-caecal)) in relation to the outcome (dead/alive at follow-up).

Survival analyses were conducted using SPSS (IBM). For each horse, the time-to-first-event (“death”) was determined from surgery (days), with the presence of death recorded as 0/1. Log-rank tests and Kaplan–Meier curves were used to determine the median survival time following surgery. Cut-off values were used from the median damage score (categorical value) for each section with the next whole integer (since a whole number scoring system is used) above this value used as the cut-point. Univariable Cox Proportional Hazards Models were constructed to assess the significance of the effects of the histological scores on the outcome, with likelihood ratio used to test the fit of the model. Hazard ratios and corresponding confidence intervals were calculated for each variable alongside the *p*-value.

Data are described as mean (95% CI) or median (IQR), with statistical significance considered at *p*-values < 0.05. Where multiple-comparison testing was performed, the Bonferroni correction was applied. A post hoc power analysis was performed using G*Power (version 3.1.9.7).

## 3. Results

Samples were obtained from 27 horses: 18 undergoing resection of a strangulating small intestinal lesion and 9 controls.

Horses undergoing surgery consisted of 13 geldings and five mares, with mean age of 15.9 +/− 4.5 years. Clinical cases comprised a mixed population of horses, representative of the caseload at the clinic, including seven Cobs, three Welsh ponies, three Thoroughbreds, one warmblood, one Irish Sports Horse, one Connemara pony, one Dales pony and one Friesian horse. Mean heart rate at presentation was 64 (±24) bpm, with mean PCV of 40 (±6) %, total plasma protein of 68 (±10) g/L and peripheral blood lactate of 2.6 (±2.1) mmol/L. The cause of intestinal strangulation was pedunculated lipoma in 11 cases, epiploic foramen entrapment in 5 cases and incarceration through a mesenteric rent in 1 case. In one horse, the cause was undetermined. Mean length of the small intestine resection was 400 (±220) cm, with ileal involvement in 5/18 cases. Jejuno-jenunal anastomosis was performed in 12 cases, jejuno-ileal anastomosis in 4 cases and jejuno-caecal anastomosis in 2 cases. The control horses consisted of five geldings, one entire male and three mares, with a mean age of 10.7 +/− 6.7 years. The breeds included four Irish Draft crosses, two Thoroughbred crosses, two Cobs and one Andalusian.

### 3.1. Histological Grading of Cases and Controls

Median histological scores were significantly higher for regions OR2 and AB2 compared to the scores of the control horses. The scores for OR1 and AB1 were not different from those of the controls (*p* = 0.56 and 0.47, respectively). The total damage scores between survivors and non-survivors were not different for regions OR1, OR2, AB1 or AB2 (Table 2).

Thrombotic vessels were not observed in any control or OR1 samples. Thrombotic vessels were identified in all other regions, in both survivors and non-survivors, with submucosal vessels being affected most frequently. Serosal fibrin was identified in one control sample and in samples from all regions investigated among the colic cases. Neither variable demonstrated a significant association with post-operative survival. The results are summarized in Table 3.

### 3.2. Survival following Surgery

Follow-up data were available for all 18 surgical cases. Five horses were alive at follow-up, with the longest reported follow-up time being 2561 days. Survival data are summarised in Table 4.

Based on the median scores for surgical cases versus controls, cut-off values for OR1 of <4, AB1 of <5, OR2 of <10 and AB2 of <11 were used to evaluate survival of horses post-surgery versus histological total damage scores. Kaplan–Meier plots for OR1, OR2, AB1 and AB2 are shown in Figure 6, Figure 7, Figure 8 and Figure 9, respectively. The median survival time for OR1 score < 4 was 66.0 (0.0, 1700.8) days; and for OR1 score ≥ 4, it was 8.0 (0.0, 17.6) days. For OR2 < 10, the median survival was 4.0 (2.6, 5.4) days; and for OR2 ≥ 10, it was 1724.0 (485.1, 2962.8) days; whereas for AB1 < 5, the median survival was 25.0 (0.0, 2218.1) days; and for AB1 ≥ 5, it was 4.0 (0.0, 52.0) days. Finally, the median survival for AB2 < 11 was 63.0 (0.0, 166.5) days; and for AB2 ≥ 11, it was 8.0 (0.0, 33.2) days. Only the model for the OR2 histological score showed a significant difference in survival (Figure 7). A post hoc power analysis based on a calculated effect size of 0.7 demonstrated that the study had a power of 0.97.

## 4. Discussion

This study reports the association of histological grading of equine small intestine with survival in a population of horses suffering from strangulating small intestinal disease where resection was performed. We did not find an association with the histological appearance of the resected tissue adjacent to the margins left in situ (OR1 and AB1) with post-operative survival. Since the tissue adjacent to the margin left in situ was not significantly different to the control tissue (using the grading system described in the present study), this suggests that the appraisal methods of tissue margins at surgery are an appropriate means of determining structurally normal margins for resection. However, despite resection to grossly visually normal tissue, eight horses (44%) did not survive beyond 10 days, suggesting that other factors, such as ultrastructural and/or biochemical differences not represented by our grading system, determine survival. Indeed, we did find that there was a relationship between the tissue damage score at the most grossly abnormal section on the oral side (OR2) and survival; albeit that a higher (worse) score was associated with longer survival, contrary to a previous study by Maescheter et al. [18].

Survival-to-hospital-discharge is a frequently used metric to describe successful surgical outcomes [4,5], although this is an artificial and misleading metric to describe post-operative survival [6]. Of greater relevance to horse welfare, and the owner, is long-term survival [5,6,21,22], although ‘long-term’ is variably interpreted as between 1 and 6 months [22], 24 months [5,21], 2 years 9 months [6] and 5 years [23]. Where long-term survival is reported, a triphasic survival curve has been described [5,6,21]. A high mortality rate is evident immediately following surgery, with a cumulative probability of survival of 0.87 by 10 days post-operatively, followed by a lower mortality up to approximately 100–120 days, then a further reduction interpreted as representative of the mortality equivalent to that of the general equine population [6]. These authors report a probability of survival for strangulating small intestinal cases of 0.7–0.8 by 100 days (depending on the exact pathology). This triphasic survival curve is not dissimilar to our findings. In our study cohort, 10/18 survived to hospital discharge, with the median time to discharge being 12.5 (7–32) days. Of these, five were still alive at follow-up, with survival beyond 66 days being associated with a low mortality rate (Table 4).

The finding that greater histological disruption was associated with increased post-operative survival is interesting, counterintuitive and difficult to explain. Ischaemic or necrotic bowel loses its bacterial barrier function, predisposing to development of peritonitis and septicaemia [24]. It is possible that a more extensive disruption to barrier function resulted in greater bacterial and endotoxin translocation, in turn generating a more rapid and florid systemic inflammatory response, resulting in more extreme or rapidly progressive clinical signs which may have expedited the decision to proceed to surgery. Thus, it may be that these horses underwent more rapid surgical correction which resulted in improved survival. Supporting the importance of a more active inflammatory response is our finding that despite no difference in histological damage scores for this region between the controls and colic cases, or between survivors and non-survivors, in region OR1 serosal fibrin was identified in 4/5 survivors compared to 3/13 non-survivors. However, it is important to recognize that this difference did not satisfy our criteria for achieving statistical significance (*p* = 0.05, Table 3).

Failure to re-establish normal small intestinal propulsive motile function during the early post-operative period can result in the destruction of the horse. The interstitial cells of Cajal (ICC) are responsible for the initiation of gastrointestinal slow-wave activity and co-ordination of propulsive motility [25]. Significant reduction in density of these cells has been associated with multiple intestinal dysmotility syndromes in humans [25] and in equine dysautonomia [26]. Fintl and co-workers, however, reported no difference in small intestinal ICC density between control horses and those undergoing resection of small intestinal strangulating lesions, indicating that other factors are more relevant [25]. We did not investigate ICC density in this study.

Both clinical and experimental reports have investigated structural disruption [13,18,19,20], biochemical changes [19,20,27,28] and cellular infiltration [17,19,20,29] to assess the severity of ischaemic GI insult. Disruption of the electron transport chain function reduced ATP, increased mucosal water content and decreased sodium and potassium content were reported after 2 h of ischaemia reperfusion in the ascending colon of ponies [27]. Our study focused on histological structural changes, as histology represents a quick, practical and relatively cheap investigation compared to other techniques used for assessing ischaemic GI damage.

In equine non-survival studies, 30 min of ischaemic insult produced mild lesions, evident upon routine histological evaluation, which progressed with both continued duration of ischaemia and once perfusion had been restored [13]. In canine ileal mucosal cells, ultrastructural degenerative changes were evident within 10 min of ischaemia, with light microscopy changes only detectable after 30 min [30]. One hour of ischaemia and reperfusion in equine jejunum produced extensive submucosal oedema, subepithelial and subserosal vesicle formation, with epithelial separation and sloughing [14]. Ultrastructural changes were less marked, with only mild intracytoplasmic vacuolation, intracellular organelles appearing within normal limits, and cell-to-cell adhesion and basement membrane integrity largely unaffected [14]. These authors concluded that the main mechanism of damage was fluid accumulation within intercellular spaces and between enterocytes and the lamina propria, resulting in the shedding of sheets of intact enterocytes. This is in broad agreement with White et al. [13], who proposed mechanical disruption secondary to increased pressure from progressive fluid and haemorrhage accumulation. These changes progressed despite restoration of perfusion and evidence of muscular activity.

Assessment of cellular infiltrates can be difficult and subjective, particularly in mild or moderate disease, considering the variability in the number, proportion and distribution of normally resident leukocytes, and the presence of gut-associated lymphoid tissue aggregates [31]. Although typical leukocyte densities, types and distribution have recently been defined in the mucosa and submucosa of the main anatomical regions of the small and large intestine in horses without evidence of GI disease [32], a diagnosis of GI tract inflammation must include an assessment of architectural changes [31]. Indeed, architectural changes have been proposed as the most critical and least subjective changes to evaluate in GI biopsies [31]. We were interested in changes throughout all layers, and particularly in grossly visually normal regions, where changes may be mild. We therefore chose to focus on structural changes in this study.

Several limitations of this study must be recognized. First, this was a study utilizing clinical cases; therefore, lesion type, medication and management prior to admission to our hospital and surgical technique could not be standardized. Equally, the duration of pathology was variable and could only be approximately determined, with no more accuracy than to within a 12 h window for a number of horses. Similarly, the length of time strangulated bowel was reperfused prior to resection differed and may have impacted on subsequent histological analysis. The location of tissue taken for analysis was based on visual appearance and obtained using techniques as consistent as possible; but, there was some inevitable variation, particularly in relation to duration from surgical reduction of incarceration until acquisition of the tissue samples. The gold standard for surgical gastrointestinal biopsies is that they are of sufficient depth, free of handling artefacts and orientated so that multiple full villus-crypt units, with associated lamina propria can be evaluated [31]. The tissue collected for this study was full-thickness, fully circumferential sections, which allowed for an assessment of all layers, from mucosal to serosal epithelia. Consistent, stable positioning in tissue cassettes during processing additionally allowed for the visualization of multiple appropriately orientated sections to be evaluated. Due to routine surgical practices during small intestinal resection, however, handling artefacts could not be avoided, and it is possible that these may have influenced the interpretation of our findings.

The grading system used was a novel one, devised to assess changes in all layers of the equine small intestine, with the total damage score generated assuming equal weighting of injury from all the layers, which may not accurately reflect biological reality. Previously published grading schemes have focused on mucosal and/or submucosal features [10,13,19], mucosal and smooth muscle cell degeneration [20], or serosal structure alone [33], and the authors are unaware of a previously reported system of objectively grading the architecture of all mural layers. Our results must therefore be interpreted in the context of application of an unvalidated assessment tool which may require further modification.

Finally, the small numbers involved in this study must be acknowledged. Whilst comparable studies have included equivalent or lower numbers of clinical cases [18,19,20], post-operative survival will be influenced by a large number of factors which cannot be controlled for in a population of client-owned horses. It would be interesting to extend this work to a larger surgical population, to see if results can be reproduced. Additionally, grading cellular infiltration would be interesting to determine if this aspect can refine the predictive power for survival.

## 5. Conclusions

In this study, we investigated the relationship of histological changes in small intestine resected from horses undergoing surgical correction of strangulating small intestinal disease with subsequent survival. Our survival analysis indicates a triphasic trajectory, in concordance with previous reports of long-term outcomes, and supports the theory that more accurate prognostic information could be derived by routinely evaluating survival 2–3 months post-operatively. Our results were unexpected in two aspects: first, histological appearance of tissue representative of normal-looking tissue left in situ is not useful in terms of predicting survival; and second, survival appears to be better in horses displaying more severe tissue disruption towards the oral end of the strangulated segments. The reasons for this remain unclear, although it could suggest that more extensive damage to the intestinal barrier function may have resulted in a more rapid progression to surgical correction, and hence improved survival. Our findings, however, must be interpreted with caution, as we employed a novel, unvalidated grading scheme in a small cohort of clinical cases. Further work could seek to expand the dataset, include additional elements in the grading scheme or assessment of biochemical markers of intestinal viability as indicators of post-operative survival.

## Figures and Tables

**Figure 1 animals-13-02715-f001:**
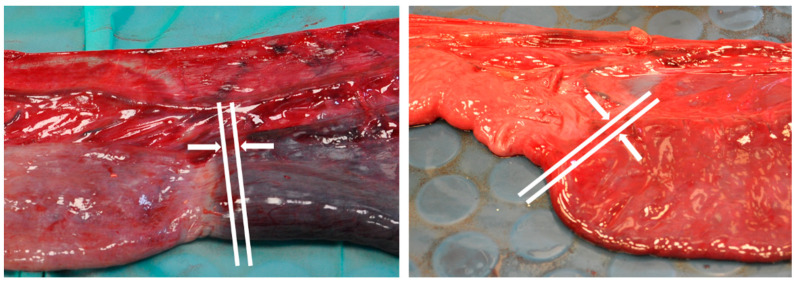
Representative images illustrating the oral (**left**) and aboral (**right**) transition zones. White lines and arrows demarcate locations from where sections OR2 and AB2 were taken.

**Figure 2 animals-13-02715-f002:**
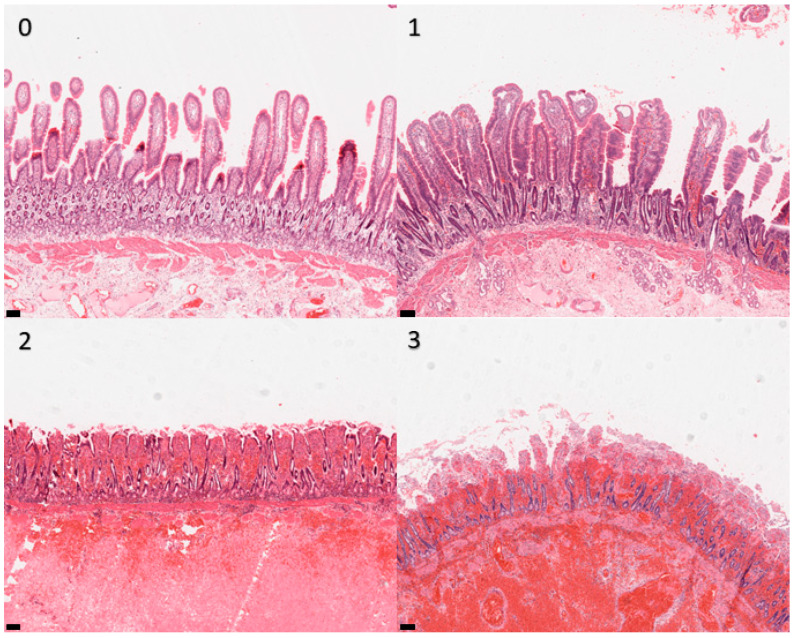
Photomicrographs of equine small intestinal sections illustrating the histological appearance of mucosa relative to the grading system; 100× total magnification, haematoxylin and eosin (for descriptors of the grades, see Table 1). Scale bar = 100 μm. 0–3: refer to the grades in the scoring system.

**Figure 3 animals-13-02715-f003:**
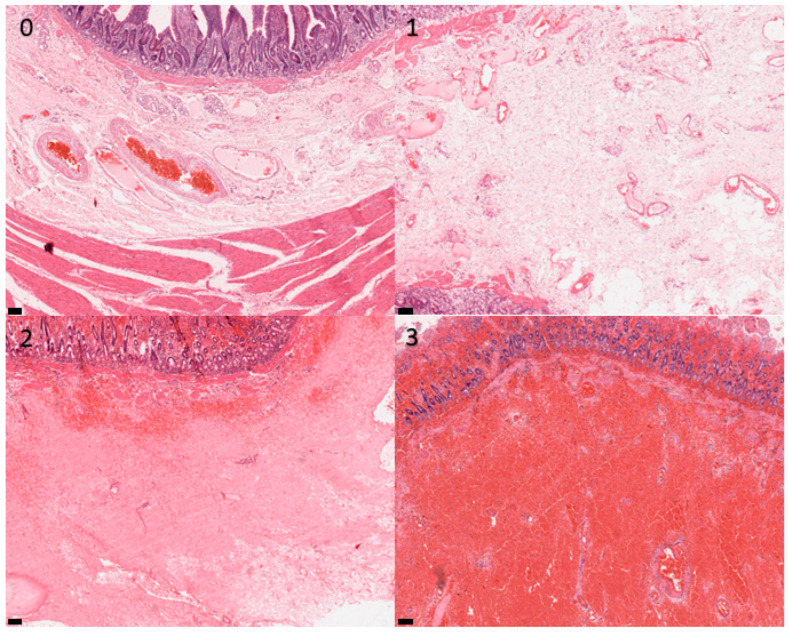
Photomicrographs of equine small intestinal sections illustrating the histological appearance of submucosa relative to the grading system; 100× total magnification, haematoxylin and eosin (for descriptors of the grades, see Table 1). Scale bar = 100 μm. 0–3: refer to the grades in the scoring system.

**Figure 4 animals-13-02715-f004:**
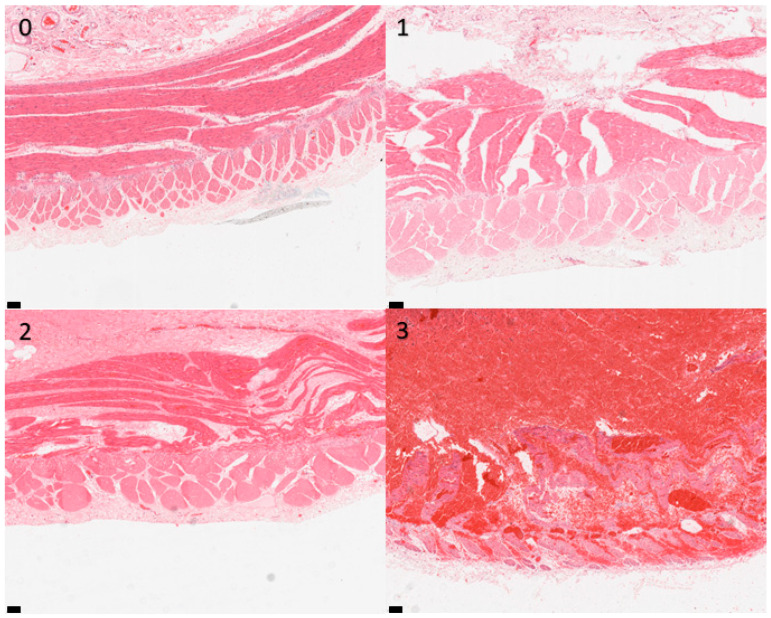
Photomicrographs of equine small intestinal sections illustrating the histological appearance of muscularis relative to the grading system; 100× total magnification, haematoxylin and eosin (for descriptors of grades 0–3, see Table 1). Scale bar = 100 μm. 0–3: refer to the grades in the scoring system.

**Figure 5 animals-13-02715-f005:**
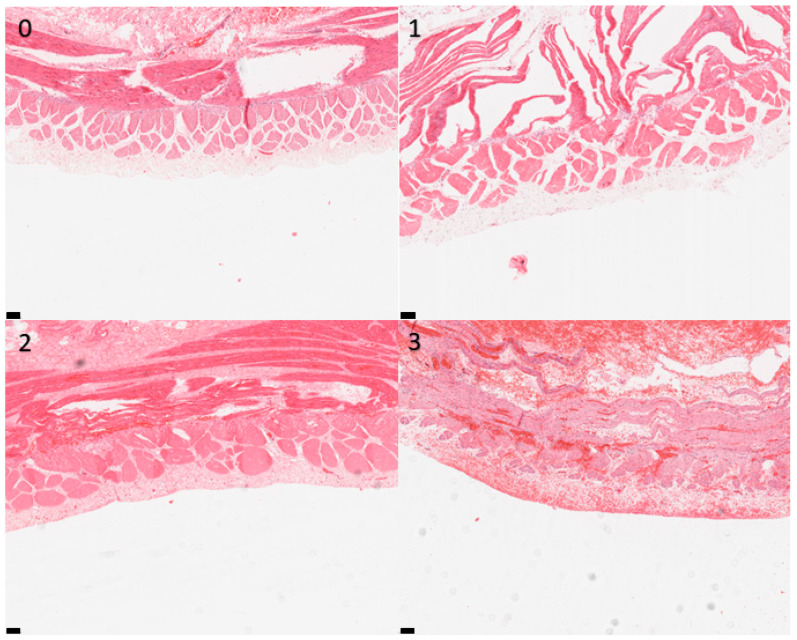
Photomicrographs of equine small intestinal sections illustrating the histological appearance of serosa relative to the grading system; 100× total magnification, haematoxylin and eosin (for descriptors of the grades, see Table 1). 0–3: refer to the grades in the scoring system.

**Figure 6 animals-13-02715-f006:**
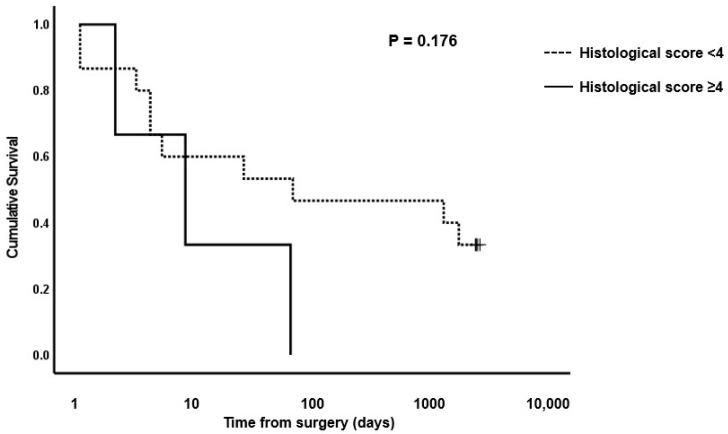
Kaplan–Meier survival plot for horses undergoing small intestinal surgical resection with histological scores of <4 or ≥4 at site OR1. The *p*-value for the log-rank test between the two score boundaries is included. *p* < 0.05 was considered significant.

**Figure 7 animals-13-02715-f007:**
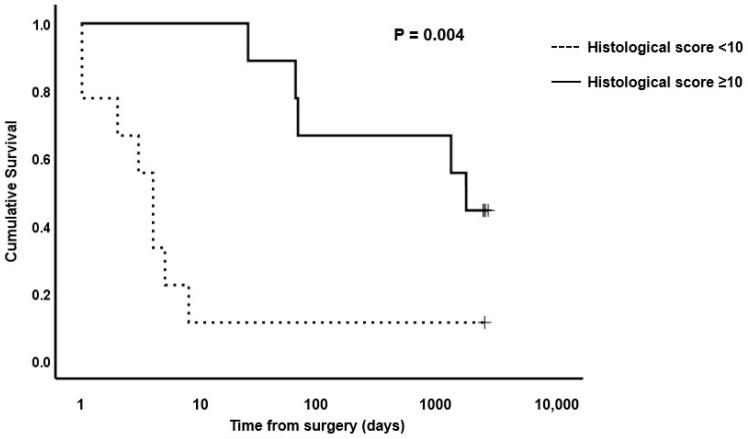
Kaplan–Meier survival plot for horses undergoing small intestinal surgical resection with histological scores of <10 or ≥10 at site OR2. The *p*-value for the log-rank test between the two score boundaries is included. *p* < 0.05 was considered significant.

**Figure 8 animals-13-02715-f008:**
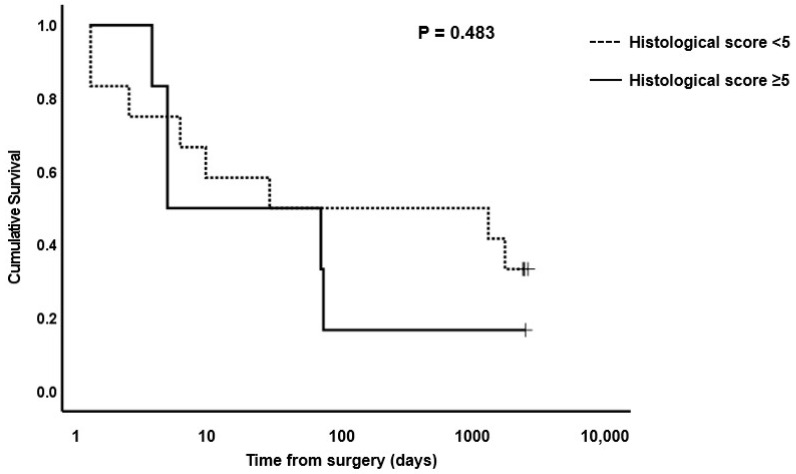
Kaplan–Meier survival plot for horses undergoing small intestinal surgical resection with histological scores of <5 or ≥5 at site AB1. The *p*-value for the log-rank test between the two score boundaries is included. *p* < 0.05 was considered significant.

**Figure 9 animals-13-02715-f009:**
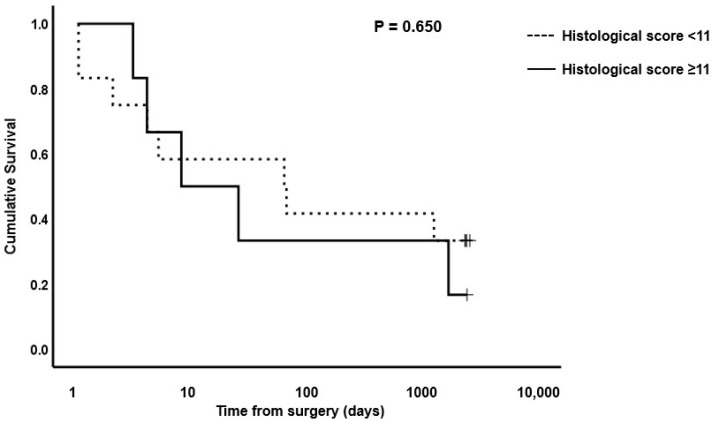
Kaplan–Meier survival plot for horses undergoing small intestinal surgical resection with histological scores of <11 or ≥11 at site AB2. The *p*-value for the log-rank test between the two score boundaries is included. *p* < 0.05 was considered significant.

**Table 1 animals-13-02715-t001:** Grading scheme applied for histological scoring of the sections. Percentages given are per five 10× fields; ^1^ GS = Gruenhagen’s space.

Grade	Mucosal Damage	Submucosal Damage	Muscularis Damage	Serosal Damage
0	Epithelium > 80% intact with GS ^1^, proprial haemorrhage and intraepithelial erythrocytes in less than 20% of villi. Proprial oedema and subepithelial clusters of macrophages might be present.	No changes to a mild degree of submucosal oedema.	No changes are observed.	Mild areas of oedema in <20%.
1	Epithelium ≥ 80% intact with GS and proprial haemorrhage, degeneration and necrosis of epithelial tips in 20–80% of villi.	Diffuse submucosal oedema with scattered small haemorrhages (in ≤20% of tissue examined).	Muscular bundles show increased intercellular space (oedema) with no haemorrhage.	Serosal oedema in ≥20% of tissue examined with no haemorrhage.
2	Epithelium 40–80% intact with multiple epithelial ulcers, necrosis and proprial haemorrhage.	Diffuse submucosal oedema with multiple, coalescing haemorrhages (20–60% of tissue examined).	Muscular bundles show increased intercellular space with haemorrhage in 20–40% of tissue examined.	Serosal oedema in 20–60% of tissue examined with occasional haemorrhage.
3	Epithelium < 40% intact with common haemorrhage and necrosis effacing vast part of the mucosa.	Common and extensive submucosal haemorrhages (>60% of tissue examined).	Muscular bundles show common haemorrhage in >40% of tissue examined.	Common (>60% of tissue examined) serosal haemorrhage.

**Table 2 animals-13-02715-t002:** Analysis of histological total damage scores of regions OR1, OR2, AB1 and AB2 in horses with strangulating small intestinal disease (cases; *n* = 18) compared to the control samples (*n* = 9), and regions OR1, OR2, AB1 and AB2 between horses still alive at follow-up (*n* = 5) and those which died or were euthanised for colic-related reasons (*n* = 13). *p* < 0.05 was considered significant.

Total Damage Score (Median (IQR))				
	Control (*n* = 9)		Cases (*n* = 18)	*p*-Value
Histological score	2 (1,3)	OR1 OR2 AB1 AB2	3 (2,3) 9.5 (9,11) 4 (2.75,6.75) 10 (8.75,11)	0.56 0.003 0.47 0.02
**Total Damage Score** **(Median (IQR))**				
**Region**	**Survivors** **(*n* = 5)**	**Non-survivors** **(*n* = 13)**		** *p* ** **-value**
OR1	3 (3,3)	3 (1,4)		0.83
OR2	11 (7,12)	9 (8,11)		0.19
AB1	3 (2,3)	4 (2,10)		0.25
AB2	10 (8,11)	10 (7,12)		0.39

**Table 3 animals-13-02715-t003:** Incidence of thrombotic vessels and serosal fibrin in regions OR1, OR2, AB1 and AB2 from horses with strangulating small intestinal disease (*n* = 18) which did (*n* = 5) or did not (*n* = 13) survive post-operatively. *p* < 0.05 was considered significant.

Thrombotic Vessels					
Region	Survivors (*n* = 5)		Non-Survivors (*n* = 13)		*p*-Value
	*n*	Location	*n*	Location	
OR1	0/5	N/A	0/13	N/A	N/A
OR2	4/5 (80%)	Serosa, submucosa	8/13 (62%)	Submucosa, muscularis, serosa	0.62
AB1	1/5 (20%)	Submucosa	3/13 (23%)	Submucosa, muscularis	1.0
AB2	2/5 (40%)	Submucosa	5/13 (38%)	Submucosa, serosa	1.0
**Serosal fibrin**					
**Region**	**Survivors (*n* = 5)**		**Non-survivors (*n* = 13)**		** *p* ** **-value**
OR1	4/5 (80%)		3/13 (23%)		0.05
OR2	3/5 (60%)		5/13 (38%)		0.61
AB1	2/5 (40%)		7/13 (54%)		1.0
AB2	2/5 (40%)		5/13 (38%)		1.0

**Table 4 animals-13-02715-t004:** Summary of the outcome data for surgical cases (*n* = 18), ranked in length of survival (days). PPID = pars pituitary intermedia dysfunction; Y = yes; N = no.

Horse No.	Alive at Follow-Up?	Time to Event (Days)	Cause of Death
1	N	0	Cardiovascular collapse
2	N	1	Post-operative haemorrhage
3	N	2	Persistent colic signs
4	N	3	Persistent post-operative reflux
5	N	4	Persistent post-operative reflux
6	N	4	Persistent post-operative reflux
7	N	5	Persistent post-operative reflux
8	N	8	Persistent colic signs
9	N	25	Collapse
10	N	63	Recurrent colic
11	N	66	Recurrent colic
12	N	1300	Laminitis/PPID
13	N	1724	Recurrent colic
14	Y	2400	
15	Y	2475	
16	Y	2478	
17	Y	2505	
18	Y	2651	

## Data Availability

Restrictions apply to the availability of these data. Data were obtained from clinical cases admitted to the University of Liverpool Philip Leverhulme Equine Hospital under Research Ethics Committee guidelines, and are available from the authors with the permission of the University of Liverpool.

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
