# Peer review of "Histological Evaluation of Resected Tissue as a Predictor of Survival in Horses with Strangulating Small Intestinal Disease"

_animals, 2023, doi:10.3390/ani13172715_

Round 1

Reviewer 1 Report

This is an interesting and well written article looking at histology at resection margins, applying a novel total damage score. However, this study has one important issue: in a study population of only 18 horses, and knowing that there are so many factors that determine survival postoperatively, I do not believe that solid conclusions can be drawn on the association with this histology score, because the influence of other factors is likely. Furthermore, not all causes of death appear to have something to do with the viability of the intestinal tissue (Collapse (from what?), Haemorrhage (from where?)), so one could question if these cases can be included in such an analysis. Also, the horses dying at 1300 and 1724 days may have died from other causes than the initial colic surgery. I am no statistician, but looking at similar studies investigating survival after colic surgery, I would think that a multivariate regression analysis would have been more suitable. Alternatively, one could stick with the current results, but then the discussion and conclusion would need to be toned down, and the emphasis and wording in the abstract and results would need to be changed accordingly.

·     Abstract: see remark above

·     Introduction:

o  Information on previous studies looking at the oral or aboral segment is lacking. Also, it is not stated which histological aspects have been evaluated.

o  Line 61 – 63: I believe that referring to other species does not really serve a purpose here

o  There is no aim or objective included

·     M&M

o  What was the study design? And how do these cases relate to the hospital population? What time frame? Were any cases excluded?

o  Line 148: at what magnification? 

o  Statistics:

§  has a power analysis been performed? 

§  There appears to be a lot of multiple testing going on. Is this corrected for?

·     Results:

o  Also see general remark

Line 235: change wording - this is not an “increase”, as it does not concern the same segment or horse

o  Line 295: I doubt that taking the study population median is the method of choice in determining cut off values in this case. It is unclear how this was calculated exactly, as it is not mentioned in the M&M section.

·     Discussion/conclusion

o  See general remark

o  One limitation that needs to be discussed, is that the TDS weighs the injury of the different intestinal layers evenly. Having seen many slides of tissue from colic horses, there are quite some samples that have complete mucosal necrosis yet with only mild damage of the other layers. Also, there may be cases that do not have severe haemorrhage but do have severe necrosis. For these cases, this score would not be suitable. 

o  Line 378 -387: ICC count is not necessarily the most important factor that has been described in the context of equine postoperative ileus. Therefore, this paragraph seems out of place.

·     Figures: the microscopic images lack a scale bar. Also, the resolution appeared quite low in the version I received. With “10x” the objective is probably meant but not the total magnification? This should be stated more clearly.

Author Response

Reviewer 1 – responses.

This is an interesting and well written article looking at histology at resection margins, applying a novel total damage score. However, this study has one important issue: in a study population of only 18 horses, and knowing that there are so many factors that determine survival postoperatively, I do not believe that solid conclusions can be drawn on the association with this histology score, because the influence of other factors is likely. Furthermore, not all causes of death appear to have something to do with the viability of the intestinal tissue (Collapse (from what?), Haemorrhage (from where?)), so one could question if these cases can be included in such an analysis. Also, the horses dying at 1300 and 1724 days may have died from other causes than the initial colic surgery. I am no statistician, but looking at similar studies investigating survival after colic surgery, I would think that a multivariate regression analysis would have been more suitable. Alternatively, one could stick with the current results, but then the discussion and conclusion would need to be toned down, and the emphasis and wording in the abstract and results would need to be changed accordingly.

Thank you for your thorough review of our manuscript and your constructive comments. We agree post-operative survival following colic surgery is a complex phenomenon. Often exact cause of death (eg collapse and haemorrhage) cannot be precisely determined as horse owners may decline permission for post-mortem examination, therefore final cause is based on clinical judgement from what information is available. However, we believe these are viable findings, despite the valid criticism of low numbers, as these are often very compromised horses with severe systemic disease that are subjected to major surgery, and the owners of these animals are interested in more than taking their horses home, but whether they will survive longer term with a quality of life and can return to a level of activity compatible with their expectations. Indeed the horses surviving to 1300 and 1724 days likely died from causes other than the colic surgeries (although one had repeated episodes of colic between surgery and euthanasia), but to survive for over 4 years post-operatively we believe justifies the decision to perform surgery. Regarding the statistical methods employed, there are a number of studies using multivariable regression analysis in relation to colic outcome (e.g. dead/alive). The advantage of time-to-event studies (survival analysis) is that they provide not only whether the event has occurred but when it occurred which regression analyses do not (these evaluate that the event has/has not occurred but not when).  Since we are interested in survival events then this approach is a more relevant statistical tool to use in this study. We hope this addresses some of your concerns, we have replied to your specific points below, and changes to the manuscript text have been highlighted.

Abstract: see remark above

Introduction:

Information on previous studies looking at the oral or aboral segment is lacking. Also, it is not stated which histological aspects have been evaluated.

We are not aware of studies which have looked at oral versus aboral margins of resected tissue; in this respect we believe this study is unique. In the added sentences (see below) stating aims and objectives, we have added that histological evaluation was intended to assess tissue architecture of full thickness samples and presence and severity of degenerative changes. We hope this satisfies the reviewer’s comment. 

Line 61 – 63: I believe that referring to other species does not really serve a purpose here

Thank you, we include this information to emphasise that this concept is not a novel one, but is frequently investigated under experimental protocols, rather than in relation to survival of clinical cases (where there are many uncontrolled, and uncontrollable variables, as the reviewer highlights in their opening paragraph). We believe it helps to set the context for our study, but are happy to exclude it if the Editor agrees.

There is no aim or objective included

Thank you, aim and objectives have now been added at the end of the Introduction, after the hypothesis (Lines 73 -81).

M&M

What was the study design? And how do these cases relate to the hospital population? What time frame? Were any cases excluded?

This was a case-control study, with samples collected over a 2 year period, this information has now been added at the start of the M&M section (Line 86). We are not quite sure what you mean by how these cases relate to the hospital population; we state that  all cases admitted with signs of colic compatible with strangulating small intestinal disease were deemed eligible for inclusion, but have rephrased lines 92-95, to clarify that if indeed strangulation of the small intestine requiring resection was identified at surgery, then these horses were recruited to the study, if that helps to clarify your query? We were able to collect samples from all horses undergoing small intestinal resection to correct a strangulating lesion over the timeframe of the study, therefore no truly eligible horses were excluded. We hope this adequately addresses your comments here, but would be pleased to know if further information is required. 

Line 148: at what magnification?

‘at 10X objective magnification’ has been added to the text to address this (Line 159).

o  Statistics:

Has a power analysis been performed?

We performed a post-hoc power analysis, and have now added this detail to the manuscript (Lines 229-230  and 317-318). An effect size of 0.7 was calculated based on central tendency values and distribution of our data set. This was used to perform a post-hoc power calculation which delivered a power of 0.97 (Faul, F., Erdfelder, E., Buchner, A., & Lang, A.-G. (2009). Statistical power analyses using G*Power 3.1: Tests for correlation and regression analyses. Behavior Research Methods, 41, 1149-1160).

There appears to be a lot of multiple testing going on. Is this corrected for?

Results:

Line 235: change wording - this is not an “increase”, as it does not concern the same segment or horse

Thank you, this has been reworded as ‘Median histological scores were significantly higher for regions OR2 and AB2..’ (Line 254).

o  Line 295: I doubt that taking the study population median is the method of choice in determining cut off values in this case. It is unclear how this was calculated exactly, as it is not mentioned in the M&M section.

Cut-off values were used from the median damage score (categorical value) for each section with the next whole integer (since a whole number scoring system is used) above this value used as the cut-point. We have added this to section 2.1 (statistical analysis) to clarify (Lines 221-223).

Discussion/conclusion

One limitation that needs to be discussed, is that the TDS weighs the injury of the different intestinal layers evenly. Having seen many slides of tissue from colic horses, there are quite some samples that have complete mucosal necrosis yet with only mild damage of the other layers. Also, there may be cases that do not have severe haemorrhage but do have severe necrosis. For these cases, this score would not be suitable.

Thank you for the comment. We took into account your (shared) vision for the scoring so, for this reason, we developed a scoring system for each intestinal layer. We did evaluate the mucosal necrosis, and have now stated this also in Table 1 to clarify this. We included haemorrhage and necrosis as distinguishing these two changes would be impossible to achieve via light microscopy (as in most of the cases they were present together) and they both reflect acute damage. We developed a TDS system to verify whether transmural infarction would be worse (or not) than if a single layer was affected.

Line 378 -387: ICC count is not necessarily the most important factor that has been described in the context of equine postoperative ileus. Therefore, this paragraph seems out of place.

Thank you for this comment. We included this information as the ICC are associated with GI motility and logically it would make sense to investigate their presence or density, considering clinical signs consistent with gut stasis are frequently the cause of post-surgical euthanasia. We did not look at ICC in this study, therefore we thought describing previous work which had found no difference in the expression of these pacemaker cells would justify this omission in our study design. We are happy to remove or relocate this paragraph if the reviewer wishes.

Figures: the microscopic images lack a scale bar. Also, the resolution appeared quite low in the version I received. With “10x” the objective is probably meant but not the total magnification? This should be stated more clearly.

Thank you for identifying this. Higher resolution images have been included in the revision, to which scale bars have been added, and total magnification has been corrected.

Reviewer 2 Report

Overall this is a very well thought out case control research project. There are globally no glaring concerns however I do have a few questions. 

1. You stated that much of the demographical data noted using the mean was all of this data really normally distributed including the length of resection, HR, lactate etc. 

2.  Can you provide referencing to where this scoring system has previously been used and if it is truly novel can you please provide more information on why this new histology scoring system is better than those that have previously been used. 

3. Lines 364-377 These are very interesting statements I agree the sicker the horse potentially the quicker it did receive anesthesia and surgical intervention is it possible that you can look back in the medical records in the history to see how long the horse was colicky for? This will help to further support your potential conclusion. 

Author Response

Reviewer 2 - responses.

Overall this is a very well thought out case control research project. There are globally no glaring concerns however I do have a few questions.

We thank the reviewer for their consideration of our manuscript and their positive comments, and address their specific concerns below. Changes to the manuscript text have been highlighted.

  1. You stated that much of the demographical data noted using the mean was all of this data really normally distributed including the length of resection, HR, lactate etc.

The first paragraph of section 2.1 (statistical analysis) states that we checked normality of data using the Shapiro-Wilk test and non-normal data (lactate) were log-transformed to achieve normality.

  1. Can you provide referencing to where this scoring system has previously been used and if it is truly novel can you please provide more information on why this new histology scoring system is better than those that have previously been used.

Thank you for this comment. We have stated in the discussion (Lines 455-459) that this scoring system was a novel one (we cannot therefore reference its previous use), and given our justification for using our scoring system in that it allows assessment off all layers of the small intestinal wall. We can not comment whether this was better than others, but we believe our system to be a very good one when dealing with full enterectomies/ post-mortem samples as we could assess completely all intestinal layers.

  1. Lines 364-377 These are very interesting statements I agree the sicker the horse potentially the quicker it did receive anesthesia and surgical intervention is it possible that you can look back in the medical records in the history to see how long the horse was colicky for? This will help to further support your potential conclusion.

Thank you, yes , we found this to be an interesting and unexpected outcome of our study, and it would indeed be of benefit if we could report times from onset of clinical signs to surgery. However, the onset time for a number of horses could not be determined more accurately than within a 12 hour window, due to when the owners had last seen the horse as ‘normal’ and when they had found the horse displaying signs of colic, we allude to this where we discuss study limitations (Lines 441-442). As our comments here are already speculation, we did not feel that any information we could provide in this context would be robust enough to be of value.

Reviewer 3 Report

Line 47: Delete “acute abdominal disease (colic)”, Colic in horses is a serious….

Line 54-55: Add “largely by subjective evaluation of gross appearance, presence of intestinal motility, surgeon experience, in combination…”

Line 61: Change “Histological changes associated with ischaemic injury of small intestine”

Line 82-83: report the pathologies included in the study

Line 90-92: specify the cm of distance “The oral and aboral ends (OR1 and AB1 respectively), avoiding the tissue that had been compressed by the application of the intestinal forceps…”

Line 122-123: describe more specifically the jejunal tract removed for each sample

Line 149: “Examination and grading were performed blind, by a single operator”, usually double blinded is better

Line 155: why did you use a new histology scoring system? report the reference from which the modification derives?

Line 227: report the pathologies for colic surgeries

Line 342-345: rephrase, it is not clear

Line 365-366: “Ischaemic or necrotic bowel loses its bacterial barrier function…”

Line 425-426: “ Firstly, this was a study utilizing clinical cases”. Or compared same pathologies

Line 47: Delete “acute abdominal disease (colic)”, Colic in horses is a serious….

Line 54-55: Add “largely by subjective evaluation of gross appearance, presence of intestinal motility, surgeon experience, in combination…”

Line 61: Change “Histological changes associated with ischaemic injury of small intestine”

Line 82-83: report the pathologies included in the study

Line 90-92: specify the cm of distance “The oral and aboral ends (OR1 and AB1 respectively), avoiding the tissue that had been compressed by the application of the intestinal forceps…”

Line 122-123: describe more specifically the jejunal tract removed for each sample

Line 149: “Examination and grading were performed blind, by a single operator”, usually double blinded is better

Line 155: why did you use a new histology scoring system? report the reference from which the modification derives?

Line 227: report the pathologies for colic surgeries

Line 342-345: rephrase, it is not clear

Line 365-366: “Ischaemic or necrotic bowel loses its bacterial barrier function…”

Line 425-426: “ Firstly, this was a study utilizing clinical cases”. Or compared same pathologies

Author Response

Reviewer 3 - responses.

Thank you for your detailed review of our manuscript. We hope we have responded adequately to your specific points below.  Changes in the manuscript text are highlighted.

Line 47: Delete “acute abdominal disease (colic)”, Colic in horses is a serious….

We have reworded this sentence as suggested (Line 47).

Line 54-55: Add “largely by subjective evaluation of gross appearance, presence of intestinal motility, surgeon experience, in combination…”

Thank you, we have reworded this following your comment to read: ‘largely by subjective evaluation of gross appearance and surgeon experience, in conjunction with…’ (Lines 54-55). We agree surgeon experience is important here, but would consider assessment of motility to be part of overall evaluation of gross appearance.

Line 61: Change “Histological changes associated with ischaemic injury of small intestine”

With respect to the reviewer, we would prefer to leave this as ischaemia-reperfusion injury as several of the studies referenced investigated changes secondary to a defined period of vascular occlusion and reperfusion, and in clinical colic cases strangulation is resolved prior to resection of affected small intestine, so a brief period of reperfusion may additionally contribute to the injury prior to acquisition of tissue samples.

Line 82-83: report the pathologies included in the study

We agree this information is important, but with respect, we believe this is better placed in the Results, rather than the M&M  section. We have here defined inclusion criteria (Horses admitted to the University of Liverpool Philip Leverhulme Equine Hospital for investigation of acute abdominal disease and taken to surgery with a presumptive diagnosis of small intestinal obstruction were considered eligible for inclusion) and then described sample collection technique. We have, however rephrased lines 92-93 to now read ‘If small intestinal strangulation requiring resection was identified at surgery, horses were included in the study and tissue samples collected’.

We have reported the pathologies, as requested, below in the Results section (Lines 243-245).

Line 90-92: specify the cm of distance “The oral and aboral ends (OR1 and AB1 respectively), avoiding the tissue that had been compressed by the application of the intestinal forceps…”

This has been reworded as follows: ‘Approximately 1 cm from the oral and aboral extremities (OR1 and AB1 respectively), avoiding tissue which had been compressed by the application of the intestinal clamps..’ (Lines 100-102).

Line 122-123: describe more specifically the jejunal tract removed for each sample

Unfortunately, a more precise description in terms of location is not possible as the control samples were taken using a small incision in the ventral midline, just large enough to admit a hand into the abdomen, with the horses in lateral recumbency. Once small intestine had been located, a short segment was exteriorised  to allow excision of a short segment, precluding accurate identification of the location along the small intestine - unless the reviewer is requesting a more detailed description as to how this segment was removed (sharp excision)?

Line 149: “Examination and grading were performed blind, by a single operator”, usually double blinded is better

We absolutely agree, but the identity of the horse and outcome was not known, or identifiable, by the grader (GR) as each slide was only identified by a number. One of the other authors (DB) had the code to then be able to match assigned grade for each section to the respective horse for further analysis. We considered this to be an acceptable method of avoiding bias in this respect.

Line 155: why did you use a new histology scoring system? report the reference from which the modification derives?

Thank you, Reviewer 2 also commented on this. We have stated in the discussion (Lines 455-459) that this scoring system was a novel one, and given our justification for using it (i.e. no previously reported scoring system described assessment of all mural layers). Other, previously reported scoring system are also referenced in this paragraph.

Line 227: report the pathologies for colic surgeries

Thank you, we have now done this as requested (Lines  243-245).

Line 342-345: rephrase, it is not clear

This has been rephrased as requested and now reads: ‘However, despite resection to grossly visually normal tissue, eight horses (44%) did not survive beyond 10 days, suggesting that other factors such as ultrastructural and/or biochemical differences, not represented by our grading system, determine survival’ (Lines  353-356). Hopefully this now conveys the message more clearly.

Line 365-366: “Ischaemic or necrotic bowel loses its bacterial barrier function…”

This has been reworded as requested (Lines 379-380).

Line 425-426: “ Firstly, this was a study utilizing clinical cases”. Or compared same pathologies

We are not quite sure what the reviewer is requesting here. We state that the study used clinical cases, with the limitations that this provides in terms of variability in lesion type (strangulating lipoma, epiploic foramen entrapment, mesenteric incarceration etc, which we have now added at the reviewer’s request). Therefore, although these were all acute small intestinal strangulating lesions, the primary pathology varied between the aetiologies listed.  We have reworded this sentence to now read: ‘Firstly, this was a study utilizing clinical cases, therefore, lesion type, medication and management prior to admission to our hospital and surgical technique could not be standardized’. We hope this addresses the reviewer’s query, but would be pleased to know if anything else is requested here.

Round 2

Reviewer 1 Report

Thank you for the revised version of the manuscript. I think the manuscript has improved in many aspects. However, some comments were not clearly addressed in the revised manuscript. As stated in my first review, the limitations of the paper could be described more thoroughly and the conclusions could be more carefully phrased considering these major limitations. Also, I would expect that the papers that have documented pre-stenotic injury in the past (Gerard, Blikslager et al. 1999, De Ceulaer, Delesalle et al. 2011) would be mentioned in the introduction of such a manuscript. If the editorial board including someone with more statistical knowledge approves of the statistical methods, I have nothing against publication.

Author Response

Thank you for the revised version of the manuscript. I think the manuscript has improved in many aspects. However, some comments were not clearly addressed in the revised manuscript. As stated in my first review, the limitations of the paper could be described more thoroughly and the conclusions could be more carefully phrased considering these major limitations.

We thank the reviewer for their additional comments. Please see our specific responses below, which we hope adequately address your concerns. These changes are again highlighted in the revised manuscript.

Lines 436-439 emphasise variability in duration of disease process and periods of reperfusion in our colic cohort as a limitation of our study.

Lines 453-454 and 458-460 add further comments on the limitations of our chosen grading scheme.

Lines 461-464 reinforce the limitation of a relatively small number of horses.

Lines 482-484 emphasise limitations interpretation of our main findings in light of the above mentioned factors.

Also, I would expect that the papers that have documented pre-stenotic injury in the past (Gerard, Blikslager et al. 1999, De Ceulaer, Delesalle et al. 2011) would be mentioned in the introduction of such a manuscript.

Thank you, these are valuable references and have now been included (numbers 19 and 20 in reference list). They are commented on in the Introduction (Lines 68-71) and referred to in the Discussion.